

# System for $\delta^{13}$C-CO$_2$ and *x*CO$_2$ analysis of discrete gas samples by cavity ring-down spectroscopy

Dane Dickinson[1], Samuel Bodé[2], Pascal Boeckx[2]

[1] Biosystems Engineering, Ghent University, Coupure Links 653, 9000 Gent, Belgium
[2] Isotope Bioscience Laboratory – ISOFYS, Ghent University, Coupure Links 653, 9000 Gent, Belgium

*Correspondence to*: Dane Dickinson (dane.dickinson@ugent.be)

**Abstract.** A method was devised for analysing small discrete gas samples (50 ml syringe) by cavity ring-down spectroscopy (CRDS). Measurements were accomplished by inletting 50 ml syringed samples into an isotopic-CO$_2$ CRDS analyser (Picarro G2131-i) between baseline readings of a standard reference air, which produced sharp peaks in the CRDS data feed.
A custom software script was developed to systematise measures and aggregate sample data. The method was successfully tested with CO$_2$ mole fractions (*x*CO$_2$) ranging from <0.1 to >20000 ppm and $\delta^{13}$C-CO$_2$ values from natural abundance up to +30000 ‰. Throughput was typically 8 to 10 samples h$^{-1}$, with 12 h$^{-1}$ possible under ideal conditions. The measurement failure rate in routine use was ca. 1 %. Calibration to correct for memory effects was performed with gravimetric gas standards ranging from 0.05 to 2109 ppm *x*CO$_2$ and $\delta^{13}$C-CO$_2$ levels varying from -27.3 to +21740 ‰. Repeatability tests
demonstrated that method precision for 50 ml samples was ca. 0.05 % in *x*CO$_2$ and 0.15 ‰ in $\delta^{13}$C-CO$_2$ for normal atmospheric CO$_2$ compositions. Long-term method consistency was tested over a 9-month period, with results showing no systematic measurement drift over time. Standardised analysis of discrete gas samples expands the scope of applications for isotopic-CO$_2$ CRDS and enhances its potential for replacing conventional isotope ratio measurement techniques. Our method involves minimal setup costs and can be readily implemented in Picarro G2131-i and G2201-i analysers or tailored for use
with other CRDS instruments and trace-gases.

## 1 Introduction

Cavity Ring-Down Spectroscopy (CRDS) is a high-sensitivity laser absorption technology becoming increasingly common for trace-gas analysis (Wang et al., 2008). As well as returning high-resolution mole fraction measurements (Crosson, 2008), CRDS is used for stable isotope analysis of CO$_2$, CH$_4$, H$_2$O, and N$_2$O (Crosson et al., 2002; Dahnke et al., 2001; Kerstel et
al., 2006; Sigrist et al., 2008). Commercial deployment of CRDS has created novel analytical possibilities with greater stability, precision, instrument portability, and a lower cost-basis compared with many traditional spectroscopic, chromatographic, and mass spectrometric techniques (Berryman et al., 2011; Hancock and Orr-Ewing, 2010; Mürtz and Hering, 2010; Picarro, 2009).





Crosson et al. (2002) provide a description of the working principles for making isotopic measurements by CRDS. Commonly used in atmospheric research, isotopic CRDS gas analysers are normally on-line instruments whereby sample gas is continuously pumped through an optical cavity. While such continuous measurement systems are useful for monitoring applications, technical adaption is necessary for routine handling of small discrete gas samples. Commercial add-on modules

are available for this purpose (McAlexander et al., 2010; Picarro, 2013), but these are unable to match the rapidity of conventional methods like gas chromatography and isotope ratio mass spectrometry (IRMS).

CRDS analysis with discrete sample throughput and handling comparable to IRMS could significantly improve a variety of empirical research. For example, simultaneous high-precision isotope ratio and mole fraction measurements from isotopic-

$CO_2$ CRDS will reduce empirical workload and increase accuracy of $CO_2$ flux partitioning calculations in soil and plant respiration experiments, especially those involving [13]C-enriched tracers (Midwood and Millard, 2011; Snell et al., 2014). However, realising these benefits requires regular batch analysis of discrete samples – existing arrangements that couple CRDS instruments directly to soil headspace chambers are generally constrained to measuring just one experiment at a time (Albanito et al., 2012; Bai et al., 2011; Midwood et al., 2008).

Berryman et al. (2011) describe a syringe sample delivery system for isotope ratio CRDS that allows small air samples (20 to 30 ml) to be analysed. In their method, the optical cavity of the CRDS analyser is flushed and completely evacuated prior to direct sample injection to ensure consistency and prevent sample-to-sample contamination. Although an important technical innovation with handling and cost advantages over IRMS, the setup is limited by slow sample turnover rates.

In this paper, we present a new method for measuring discrete syringed gas samples (50 ml) by CRDS. Like Berryman et al. (2011), this method was conceived for isotopic-$CO_2$ CRDS to provide $\delta^{13}$C-$CO_2$ and $CO_2$ mole fraction analysis in soil respiration studies, but remains general enough to be used in other contexts and adjusted for other gas species. Instead of evacuating the cavity prior to sample introduction, our process intersperses samples against background measurements of a

fixed reference air and post-corrects for bias in the measurements. This results in considerably faster throughput for typical atmospheric samples than the method of Berryman et al. (2011). Additionally, with precision and discrete sample measurement rates comparable to automated continuous-flow IRMS, this method further advances CRDS as an attractive alternative for trace-gas isotope analysis.

## 2 Materials and methods

### 2.1 Analyser and sampling system

The CRDS instrument adapted for discrete sample measurement was a Picarro G2131-i isotopic-$CO_2$ gas analyser (Picarro Inc., Santa Clara, CA, USA). Detailed description of the operation and spectroscopy of the G2131-i and predecessor units



can be found in Dickinson et al. (2017), Hoffnagle (2015), Rella (2010a, 2010b, 2010c), and Wahl et al. (2006). In brief, sample air is circulated through a high-reflectivity optical cavity (35 cm$^3$) at a flow rate of ca. 25 ml min$^{-1}$. Internal controls maintain the cavity at 318.150 ± 0.002 K and 18.67 ± 0.02 kPa. Spectroscopic ring-down time constants are measured across spectral bands of $^{12}C^{16}O_2$ and $^{13}C^{16}O_2$ to determine optical absorption peak heights, which are computed into $^{13}C/^{12}C$ isotope

ratio and $CO_2$ mole fraction data (Hoffnagle, 2015). Spectral lines of $^{12}C^1H_4$ and $^1H_2^{16}O$ are also measured for correcting direct and indirect spectral interferences from $H_2O$ and $CH_4$ on the $CO_2$ bands. The normal measurement range for the G2131-i is set at 380 to 2000 ppm $xCO_2$ and natural abundance to +5000 ‰ in $\delta^{13}C$-$CO_2$ (Picarro, 2011).

All measurements made by the G2131-i are continually recorded at a rate of ca. 0.8 Hz; specific data must be extracted from

log files for further treatment. Although discrete sample measurement is thus possible without special provision – by inletting the G2131-i with 200 to 300 ml of sample from a gasbag or chamber and retrieving the relevant data (Picarro, 2012) – such procedure is time inefficient and prone to errors from operator inconsistency. Furthermore, in many research settings it is impractical or impossible to gather such large samples (e.g. headspace chamber analyses). By instead applying a controlled procedure for inletting smaller volumes and software to automatically process the raw data, a more feasible

method of discrete sample measurement was created.

A schematic of the measurement set-up is shown in Fig. 1. The system was simple in construction and concept: Hermetic sample collection and delivery was achieved by high-quality gas-tight syringe with push-button valve and Luer lock fitting (50 ml, SGE Analytical Sci., Australia). A low permeability multi-layer foil gasbag (27 L Plastigas, Linde AG, Germany)

functioned as a reservoir for a standard reference air, which was analysed between individual samples so as to give a 'baseline' for accurate data delineation. The large, non-pressurised volume of the reservoir meant pressure induced mixing and back-flow risks were excluded, and allowed prolonged operation (>15 h) before refilling. Gas-proof fluorinated-ethylene-propylene (FEP) tubing (Rotilabo, Carl Roth GmbH, Germany) and Luer lock 3-way valves completed the set-up. All permanent tube fittings and joins were adhered with Loctite 406/770 (Henkel AG, Germany) to ensure robustness and

prevent leakage. The FEP tubing between the syringe sample inlet point and the CRDS inlet port (Fig. 1) was minimised (⅛″ OD × 44 cm) to reduce mixing and lag time between sample delivery and measurement.

## 2.2 Sample measurement

The G2131-i and discrete sample measurement system were installed in an environmentally controlled laboratory (20 °C) to ensure stable operation. Syringed sample measurement was performed as follows: After insturment start-up and

commencement of normal function, reference air measurement was initiated to establish stable baselines of $xCO_2$ and $\delta^{13}C$-$CO_2$. When a sample was ready for analysis, the syringe was connected to the sample inlet point (Fig. 1), and the 3-way valve manually actuated to stop the flow of reference air and supply the sample directly to the analyser. Upon opening the syringe valve the gas sample was automatically drawn into the G2131-i, causing steady, unassisted collapse of the syringe



plunger. Sample evacuation was completed in ca. 2.5 min, after which the sample inlet point valve was manually reset and reference air intake resumed. Once $CO_2$ and $\delta^{13}C$-$CO_2$ readings had returned to initial baseline levels (thereby safeguarding against sample-to-sample carryover), the process was repeated for the next sample. In this way, reference air readings were punctuated by syringe samples to create 'peaks' in the data output with a sample-to-sample time of ca. 5 min (Fig. 2). The

gas aliquot size for all measurements was nominally 50 ml NTP. (Analysis of smaller amounts may be possible but 50 ml was assessed as a minimum for reliable operation. Samples larger than 50 ml would be easily handled, although adjustment of peak truncation parameters and re-calibration may be necessary for accurate performance – see below and Sect. 2.3.)

To achieve unambiguous sample peak identification, distinction in $CO_2$ was required between reference air and sample. In

practice this meant a relative change of ca. 2 % in $xCO_2$ or ca. 5 ‰ in $\delta^{13}C$-$CO_2$. However, very large differences resulted in slower sample throughput (see Sect. 3.1). Best results were obtained using reference air that was similar to samples in $CO_2$ mole fraction but contrasting in $\delta^{13}C$-$CO_2$ (e.g. 15 ‰ difference). In this work, dry standard air with 496 ppm $xCO_2$ and - 36.1 ‰ $\delta^{13}C$-$CO_2$ was used as the reference for all formal measurements (NA1, Table 1).

In the course of system testing, a basic algorithm for peak recognition and data extraction was developed. Due to the difficulty of real-time data flagging on the G2131-i, specific events and timings in the measurement process were used to isolate samples (Fig. 3a). Prior to introduction of a sample, a reference air baseline was recorded for 30 s and averaged. Sample detection (trigger) then occurred when $xCO_2$ or $\delta^{13}C$-$CO_2$ values deviated from the baseline beyond a fixed threshold (default: 0.5 % of $xCO_2$ or 2 ‰ in $\delta^{13}C$-$CO_2$). The sample end (detrigger) was detected when measures returned halfway to

baseline values (Fig. 3b). By truncating the sample peak data +80 s from the trigger and -29 s from the detrigger, ca. 30 s of representative measurement data was obtained for each sample (Fig. 3a).

Data processing was handled by a custom computer script that we composed for use within the built-in software architecture of the G2131-i. Running through the Picarro 'Coordinator' application program, our script regulated measurements to ensure

correct timing for user introduction of samples, detected and extracted sample peak data, monitored reference air values, and filtered problem measures. For each sample, the script computed means and standard deviations (SDs) of all data elements reported by the G2131-i (i.e. $xH_2O$ and $xCH_4$ values along with $xCO_2$ and $\delta^{13}C$-$CO_2$). These statistics were then compiled along with corresponding baseline measures, assigned sample descriptors, and finally outputted into concise results files (see example in the Supplement).


In addition to the G2131-i analyser, our method was successfully trialled on a sister CRDS instrument (Picarro G2201-i). The G2201-i differs from the G2131-i only in additionally measuring $^{13}C^1H_4$ to give $\delta^{13}C$-$CH_4$ data (Picarro, 2015). To assist method adoption, we supply software scripts customised for each instrument (Supplement). The scripts include provision for user-adjustment of peak identification and truncation parameters to suit individual set-ups.



## 2.3 Measurement calibration

As discussed in studies by Gkinis et al. (2011) and Stowasser et al. (2012), stepwise changes to the inlet gas composition (as occur with discrete samples) do not give rise to correspondingly abrupt jumps in CRDS measurements, and instead result in sigmoid-shaped steps in the data (Fig. 3b). These smoothed transitions are the combined result of (i) the rate of gas

replenishment in the optical cavity (Stowasser et al., 2014), (ii) partial mixing (turbulence and diffusion) of gas compositions downstream of the sample inlet (Gkinis et al., 2011), and (iii) molecular sorption and desorption on internal surfaces of the cavity and inlet tubes (Friedrichs et al., 2010). Although 'response times' of CRDS instruments typically range 1 to 3 min (Picarro, 2011; Sumner et al., 2011), the actual time required for an optical cavity to completely transition to a new gas composition can be substantially longer. In testing the G2131-i, we observed remnants of previous gases persisting with

asymptotical decline for as long as 40 min following very large shifts in $CO_2$ composition (e.g. $|\Delta xCO_2|$ >10000 ppm or $|\Delta\delta^{13}C\text{-}CO_2|$ >5000 ‰). While the error caused by the residual gases may sometimes be relatively trivial, all measurements that occur prior to the cavity attaining equilibrium will experience these 'memory effects'.

In the case of our 50 ml syringe samples, memory effects were clearly present, as evidenced by the asymptotic curvature in

the data peaks (Fig. 2). This meant that reported measures of syringe samples were biased towards reference air compared to 'true' values that would be determined from measurements of indeterminately large sample volumes and monitoring for asymptotic closure. Other researchers have mitigated memory effects by evacuating the optical cavity before sample injection (Berryman et al., 2011), or through several replicate measurements (Gupta et al., 2009; Leffler and Welker, 2013). In this work however, we elected to post-correct for reference air contamination by calibrating our measurement method

with bottled gas standards. More specifically, we compared discrete sample measurements of gas standards against measures of the same standards directly inlet to the G2131-i for prolonged periods (>1 h). Importantly, no attempt was made to calibrate the syringe measurements directly against the gravimetric values of the standards – we were only concerned with isolating the bias associated with syringe sampling and not with any inaccuracies internal to the instrument spectroscopy (calibration of which should be undertaken separately; see for instance Dickinson et al., 2017). In this way we prevented

convolution of errors that might have occurred if combining multiple layers of corrections into one step.

To this end, seven gravimetric gas standards with wide variation in $CO_2$ composition (0.05 to 2109 ppm $xCO_2$, -27.3 to +21740 ‰ $\delta^{13}C\text{-}CO_2$) were used as fixed source calibrants (see Table 1; exact compositions detailed in Dickinson et al., 2017). Direct measurements were performed by inletting the bottled standards to the G2131-i for more than one hour to

ensure the absence of memory effects before taking formal measures for 10 min (ca. 460 data points; averages reported in Table 1). Next, 50 ml syringe samples of the standards were taken directly from bottles (syringe was pre-flushed several times to preclude contamination) and measured using our method as outlined (ca. 8 samples of each standard, for 56 measures in total – dataset in the Supplement). Before further analysis, due to the high $^{13}C$-enrichment in several gas




standards, all reported $CO_2$ data were adjusted for accuracy by the formulae in Dickinson et al. (2017), thereby eliminating unaddressed interferences and calculation abnormalities in the internal spectroscopy of the G2131-i.

The relationship between syringe and bottle measurements was established by recognising that the data peaks generated by
syringe samples could be approximated by generalised logistic curves (Fig. 3b; also Gkinis et al., 2011). From this, together with a constant aliquot size for all syringe measures, we were able to predict a simple linear scaling of syringe values:

$$syringe = base + (bottle - base)/K \qquad (1)$$

where *syringe* refers to the measurement value obtained for a syringe sample of a gas standard, *base* to the baseline measurement of reference air prior to sample introduction, *bottle* to the direct measurement of the same standard, and *K* is a
dimensionless empirical constant.

While all $CO_2$ data elements reported by the G2131-i exhibited reasonably similar sample peak geometry, the empirical constants for $^{12}CO_2$ and $^{13}CO_2$ were expected to differ due to (de)sorption and diffusion induced isotope fractionation during sample filling of the optical cavity. Further, theoretical gas mixing considerations entailed Eq. (1) would not consistently
hold for $^{13}C/^{12}C$ isotope ratio data ($R_{CO_2}$) where a simultaneous change in total-$xCO_2$ also occurred. Consequently, only $x^{12}CO_2$ and $x^{13}CO_2$ data were explicitly calibrated, with $R_{CO_2}$ being subsequently recalculated. (Moreover, only the dry mole fraction data of $^{12}CO_2$ and $^{13}CO_2$ were used due to the high likelihood of different transition equalisation rates for $CO_2$ and $H_2O$. For explanation of dry and wet mole fraction data see: Hoffnagle, 2015; Rella, 2010a; Rella et al., 2013.) Accordingly, the following correction formulae were derived from Eq. (1):

$$x^{12}CO_2(corrected) = x^{12}CO_2(base) + [x^{12}CO_2(syringe) - x^{12}CO_2(base)] \cdot K_{C12} \qquad (2)$$

$$x^{13}CO_2(corrected) = x^{13}CO_2(base) + [x^{13}CO_2(syringe) - x^{13}CO_2(base)] \cdot K_{C13} \qquad (3)$$

Total-$xCO_2$, $R_{CO_2}$, and $\delta^{13}C\text{-}CO_2$ data were then determined from the resulting corrected values of $x^{12}CO_2$ and $x^{13}CO_2$:

$$xCO_2 = x^{12}CO_2(corrected) + x^{13}CO_2(corrected) \qquad (4)$$

$$R_{CO_2} = \frac{x^{13}CO_2(corrected)}{x^{12}CO_2(corrected)} \qquad (5)$$

$$\delta^{13}C\text{-}CO_2 = \left[\left(\frac{R_{CO_2}}{R_{VPDB}}\right) - 1\right] \cdot 1000\ ‰ \qquad (6)$$

The correction constants, $K_{C12}$ and $K_{C13}$, were found through weighted least squares analysis (WLS) of Eqs. (2) and (3) with syringe and bottle measurements of gas standards as input data (i.e. reverse regression of Eq. 1; bottle measures substituting for the left-hand-sides of Eqs. 2 and 3). To increase statistical power, $R_{CO_2}$ and total-$xCO_2$ data from bottle measurements were also incorporated into the analysis with Eqs. (4) and (5), thereby forming an extended optimisation problem (n = 216).
In a similar vein to the WLS approach used by both Dickinson et al. (2017) and Stowasser et al. (2014) for calibrating CRDS





measures, residuals weights were taken as the reciprocals of the individual summed variances resulting from the SDs of each syringe sample and bottle measurement (see Supplement and Table 1). The WLS solution was determined in R (version 3.2.1; R Core Team, 2015) by general purpose optimisation using the L-BFGS-B algorithm (Zhu et al., 1997) to yield the best-fit correction constants for all available $CO_2$ mole fraction and $^{13}C/^{12}C$ isotope ratio data.

**2.4 Precision and consistency tests**

The gas standards used for compensating memory effects in syringe sample measurements covered a wide span of $CO_2$ mole fractions and very high $\delta^{13}C$-$CO_2$ values. While this was necessary for ensuring calibration accuracy and applicability, because several of the standards contained $CO_2$ compositions beyond the normal operating range of the G2131-i, those data were inappropriate for drawing conclusions about measurement precision.

Precision of method was therefore evaluated by replicate measurements of a systematic set of $CO_2$ mixtures that better conformed to G2131-i specifications. Using gas standards as blending sources (Table 1; Dickinson et al., 2017), 20 unique mixes with varied $CO_2$ mole fractions (ca. 300, 600, 1000, 1500, 2000 ppm) and $\delta^{13}C$-$CO_2$ values (ca. -30, +800, +1750, +2700, +3600 ‰) were prepared into multi-layer foil gasbags (1000 ml Supel Inert, Sigma Aldrich). (The set of mixtures formed an orthogonal array of cross combinations of $x$$CO_2$ and $\delta^{13}C$-$CO_2$; cf. Fig. S1 in the Supplement.) Each mix was sampled and measured with the syringe method three times in succession, and results analysed for inter- and intra-measurement variability.

Long-term consistency and reliability of our syringe method was assessed by periodic analysis of a standard air (NA2, Table 1) during the course of 9 months of routine instrument use. More than 200 measurements were conducted and results examined for precision and drift.

**3 Results and discussion**

**3.1 System operation**

Though somewhat labour intensive and requiring continual operator presence, the syringe sample measurement process was uncomplicated, reliable, and economical. Sample handling and CRDS operation was non-specialist in comparison to conventional IRMS. The method was flexible to $CO_2$ composition, successfully handling samples <0.1 to >20000 ppm $x$$CO_2$ and -100 to +30000 ‰ $\delta^{13}C$-$CO_2$. The only significant methodological constraint observed was a reduction in sample turnover rate for compositions greater than either 3000 ppm $x$$CO_2$ or +4000 ‰ $\delta^{13}C$-$CO_2$. This was because post-sample reference air measures took longer to return to pre-sample baselines due to memory effects, thereby extending the inter-sample period. Keeping $CO_2$ levels within G2131-i specifications resulted in a throughput of 8 to 10 samples h$^{-1}$. Best measurement rates of 12 to 13 samples h$^{-1}$ occurred when sample $CO_2$ compositions neighboured the reference air (e.g.



within ca. 100 ppm $xCO_2$ and ca. 20 ‰ $\delta^{13}$C-$CO_2$ of reference). These throughput rates are at least a 2-fold improvement over both the method of Berryman et al. (2011) and specialty peripheral devices (Picarro, 2013).

Following initial development, the syringe method was incorporated into our general laboratory practices and during the first
year of implementation more than 10000 samples were measured. Despite intense instrument usage, we noticed no changes or adverse impacts on G2131-i function, although increased external pressure variations caused by frequent syringe evacuations may conceivably reduce mechanical lifetimes of optical cavity pressure control valves. Failures occurred in ca. 1 % of measurements, principally due to operator mistakes, but occasionally because of leakage in sample inlet valve, syringe fault, or complications from the peak identification algorithm for samples very similar to the reference air (see Sect. 2.2).
Very rarely, minor instabilities in reference air readings caused false peak detections and baseline return problems, but such instances were usually identified by the software script and internally resolved.

Durability of the gas-tight syringes used for sample delivery was excellent, although regular monitoring and maintenance was important to ensure smooth sample evacuation during the measurement process. Excessive plunger friction led to
significant 'jumpiness' in syringe collapse, which manifested as small pressure fluctuations to the optical cavity and increased measurement noise (evidenced by larger reported SDs). Careful cleaning and exact silicone lubrication was carried out every few hundred samples to ensure uniform plunger operation and prolongation of syringe life. Syringe push-button and sample inlet point valves also required periodic attention and were replaced as necessary to pre-empt leaks and breakages.

**3.2 Correction of memory effects**

From rearranging Eq. (1), the discrepancy between syringe and bottle measurements (syringe bias) was predicted to be proportional to the difference of the syringe value and reference air baseline (sample peak height):

$$(syringe - bottle) = (syringe - base) \cdot (1 - K) \tag{7}$$

Comparing the actual syringe sample and bottle measurements of gas standards, we observed systematic memory effect bias
that was indeed consistent with this postulated relationship (Fig. 4). WLS across all $CO_2$ data yielded estimates of 1.00341 for $K_{C12}$ and 1.00440 for $K_{C13}$, with a coefficient of determination ($r^2$) of 0.84 (weighted residuals) for the complete correction model. Standard errors for $K_{C12}$ and $K_{C13}$ estimates were respectively 0.00017 and 0.00014 (see confidence intervals in Fig. 4). The Pearson's correlation coefficient between $K_{C12}$ and $K_{C13}$ estimates was 0.26. The observed divergence in correction constants for $^{12}CO_2$ and $^{13}CO_2$ was statistically significant (t-test: P < 0.0001) with a larger memory
effect present in $^{13}CO_2$ measurements. This result corroborates the expectation of isotope fractionation occurring during gas equalisation in the CRDS optical cavity, prospectively due to surface (de)sorption and diffusion phenomena.





Having determined $K_{C12}$ and $K_{C13}$, syringe $CO_2$ measurements can be adjusted for bias with Eqs. (2)–(6). Accuracy of these corrections is very good: The standard errors on $K_{C12}$ and $K_{C13}$ add uncertainty to $xCO_2$ and $\delta^{13}C$-$CO_2$ data of less than 0.02 % of the difference between the sample and baseline values. For typical atmospheric samples, this additional source of error is entirely negligible compared to the uncertainty deriving from measurement precision and gas standard analytical accuracy.

While the correction coefficients ($K_{C12}$ and $K_{C13}$) found in this work are unique to our sampling equipment and G2131-i analyser, the equivalent calibration may be easily performed on replica setups. We provide a generic spreadsheet to post-correct syringe sample $CO_2$ data for any values of $K_{C12}$ and $K_{C13}$, and a template for simultaneously applying the syringe correction with the spectroscopic calibration strategy of Dickinson et al. (2017) for $^{13}C$-enriched samples (Supplement). Although our work only addresses memory effect bias in $CO_2$ data, we are confident the same strategy (Eq. 1) is straightforwardly applicable to other gas species (and isotopes) that can be similarly analysed by syringed samples and CRDS (e.g. $CH_4$, $H_2O$, $N_2O$).

### 3.3 Measurement precision and consistency

Precision of CRDS data can be evaluated in several ways: The SD of a moving average is a common approach in continuous analyses of a dynamic source (e.g. the ambient atmosphere; Zellweger et al., 2016) while measures of homogenous gas sources can be assessed by the SD of replicated samples (e.g. Wang et al., 2013) or by the SD of aggregated data in a single long-duration measurement (e.g. ≥10 min; as in Sect. 2.3 for bottle measurements, also Pang et al., 2016; and Stowasser et al., 2014). For our case of 50 ml syringe samples, replicate tests provided a detailed account of precision throughout the normal operational $CO_2$ range of the G2131-i, quantified in terms of both internal variation in individual sample analyses (i.e. intra-sample SD of the ca. 30 s of CRDS data comprising each measure, see Sect. 2.2) and the statistical dispersion of repeated samples (i.e. inter-sample SD).

Figure 5 shows inter- and intra-sample SDs and relative SDs for $^{12}CO_2$ and $^{13}CO_2$ mole fraction data (complete dataset in the Supplement). The SDs of both species were generally proportional to their measured values and unaffected by $\delta^{13}C$-$CO_2$ level (i.e. precision in $^{12}CO_2$ and $^{13}CO_2$ measurements were mutually independent). Relative SDs for both isotopolouges remained near constant at ≤0.05 % across the tested ranges however (Fig. 5c, d). Notably, the majority of intra-sample SDs for both $x^{12}CO_2$ and $x^{13}CO_2$ data were found to be in general agreement with counterpart inter-sample SDs (see trendlines in Fig. 5). This means that the SDs reported by our software script for $^{12}CO_2$ and $^{13}CO_2$ mole fractions in individual syringe sample measures will reasonably approximate the expected precision for replicated measurements of those samples.

On the other hand, inter- and intra-sample SDs in $^{13}C/^{12}C$ isotope ratio data were dependent on the $\delta^{13}C$-$CO_2$ level and $CO_2$ mole fraction, increasing with higher $\delta^{13}C$-$CO_2$ and lower $xCO_2$ (see Fig. S1a, b in the Supplement). The relative SDs of isotope ratio measurements were unaffected by $\delta^{13}C$-$CO_2$ level but steadily decreased with increasing $xCO_2$ – declining from





between 0.07 and 0.04 % at 300 ppm $x\mathrm{CO_2}$ to between 0.03 and 0.015 % at 2000 ppm (Fig. S1d). One exception was at natural abundance $\delta^{13}$C-CO$_2$ levels (ca. -30 ‰) where inter-sample relative SDs of $R_{\mathrm{CO_2}}$ were steady at ca. 0.015 % (i.e. 0.15 ‰) across the tested $x\mathrm{CO_2}$ range (Fig. S1b). Slightly opposing CO$_2$ mole fraction data, intra-sample SDs of isotope ratio data were almost always greater than corresponding inter-sample SDs, which largely reflects the summation of variance from the

$^{12}$CO$_2$ and $^{13}$CO$_2$ spectral measurements used to generate the $^{13}$C/$^{12}$C ratios. Nevertheless, as with $^{12}$CO$_2$ and $^{13}$CO$_2$, the SD reported for $\delta^{13}$C-CO$_2$ in individual syringe sample measures may be used as a conservative proxy of $\delta^{13}$C-CO$_2$ replicate precision.

Consistency of the syringed sample method was established by long-term repeated analysis of a standard air (NA2, Table 1).

Figure 6 shows $x^{12}$CO$_2$ and $\delta^{13}$C-CO$_2$ data from 200 measurements covering a 9-month period (dataset available in the Supplement). These measures averaged 1024.18 ppm in $x^{12}$CO$_2$ and -27.35 ‰ in $\delta^{13}$C-CO$_2$ with respective SDs of 0.50 ppm and 0.33 ‰. The latter SD is slightly larger than the inter-sample SDs found in replicate measure testing, likely indicating the presence of instrument drift in the data in addition to random errors from repeated sampling. However, moving-means (red lines in Fig. 6) show there was neither a sustained time-series trend nor method discontinuity, and imply that reasonable

measurement accuracy is possible without perpetual calibration against gas standards (cf. syringe sample measures of NA2 against the direct bottle measurement; Fig. 6). The mean of intra-sample SDs in the measures was 0.42 ppm for $x^{12}$CO$_2$ and 0.35 ‰ for $\delta^{13}$C-CO$_2$, both corresponding well to the aforementioned SDs of all measurements and the intra-sample SDs in the replicate tests. This consistency further supports our proposition that a single syringe measure and its intra-sample SD can deliver a similar (although inherently less reliable) statistical estimate to one generated through multiple sample

measurements, potentially making replicate CRDS analyses unnecessary in research contexts where statistical uncertainty is not a critical consideration.

In sum, despite the short CRDS analysis period for a syringe sample (~30 s), and limited number of replicates in performance testing, achieved measurement precision was excellent. With our system and G2131-i analyser, replicate

sample SDs of ≤0.05 % may be expected for $^{12}$CO$_2$ and $^{13}$CO$_2$ mole fraction data at atmospheric CO$_2$ concentrations, while typical resolution in $\delta^{13}$C-CO$_2$ measurements will be ca. 0.15 ‰ at natural abundance. Moreover, to a first approximation, similar precisions can be obtained from intra-sample SDs of single syringe sample measures. Our results improve upon the 0.3 % ($x\mathrm{CO_2}$) and 0.3 ‰ ($\delta^{13}$C-CO$_2$) precision attained by the system of Berryman et al. (2011), although this is partly due to the enhanced spectroscopic sensitivity of the G2131-i compared to the older G1101-i analyser used in their study.

Additionally, our method delivers precision in $\delta^{13}$C-CO$_2$ almost equal to the Picarro SSIM2 discrete sample peripheral device (0.11 ‰; Picarro, 2013) and traditional continuous-flow IRMS (~0.1 ‰; Prosser et al., 1991), which, by contrast, are single-purpose instruments that do not also report accurate CO$_2$ mole fraction measurements. And finally, although finer measurement resolution is possible with CRDS (e.g. by analysing larger gas aliquots or with continuous sampling; Pang et al., 2016), the uncertainties deriving from the precision of our discrete sample measures will be, in many cases, no worse



than the typical tolerances on gravimetric gas standards used for instrument calibrations (cf. Brewer et al., 2014; Dickinson et al., 2017). In such contexts, applying our method for isotopic and mole fraction analyses of trace-gases should generally not result in significantly poorer absolute accuracy compared to other sampling techniques (i.e. uncertainties on gas standards, rather than measurement precision, may limit overall accuracy).

**4 Conclusions and outlook**

Discrete sample analysis of trace-gases by CRDS is possible through basic instrument adaptation. We have set forth a scheme for $xCO_2$ and $\delta^{13}C\text{-}CO_2$ determination of 50 ml syringed samples on a Picarro G2131-i isotopic-$CO_2$ analyser. With software to manage the measurement process and compute results data, our method offers substantially faster analysis of small gas volumes with equal or better precision than comparable systems. Memory effects present in syringe sample

measurements can be accurately compensated by calibration against large-volume measures of gravimetric gas standards.

Although CRDS is gaining scientific acceptance for isotopic-$CO_2$ measurement, so far the technology has not seriously challenged IRMS in discrete gas sample analysis, despite lower running and capital costs, simpler operation, less measurement drift, and the added benefit of providing accurate $xCO_2$ data concurrently with $\delta^{13}C\text{-}CO_2$. In achieving similar

precision and sample throughput to IRMS, our syringe sample method helps position CRDS as a tenable competitor for isotopic analysis of discrete samples.

The chief disadvantages of our process compared to IRMS for isotopic-$CO_2$ analysis are a narrower operational $xCO_2$ range, a comparatively larger sample requirement (50 ml NTP), and higher labour demands. Method refinements that integrate

automatic syringes and valve systems may greatly ease operator workload however, and the development of smaller optical cavities could reduce the gas volume needed for discrete sample analysis on future CRDS instruments (e.g. Stowasser et al., 2014).

This system can be applied with any Picarro G2131-i or G2201-i CRDS analyser, though calibration and tuning of

parameters in the software script may be necessary to account for variations in set-up, sample volume (and pressure), and reference air composition. Implementation on other CRDS instruments and conversion for measurements of other trace-gases are anticipated with only minor software amendments.

**Supplement items**

- Fig. S1. Precision in syringe sample $^{13}C/^{12}C$ isotope ratio data
- Discrete sampling software scripts for Picarro G2131i and G2201i analysers
- Example_discrete_sample_data_output.csv



- Measurement_data.xlsx
- Templates for bias correction (2)

*Acknowledgements.*

We are grateful to our colleagues Stijn Vandevoorde, Hannes De Schepper, and Katja Van Nieuland at ISOFYS (Ghent
University, Belgium) for their assistance with instrument operation and numerous sample measurements. We also thank Lei
Liu (CREAF-CSIC, Barcelona, Spain) for testing our method on a Picarro G2201-i CRDS unit and for providing feedback
on method efficacy. Renato Winkler from Picarro Inc. aided this work with his useful advice on developing software scripts
for the G2131-i analyser.

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



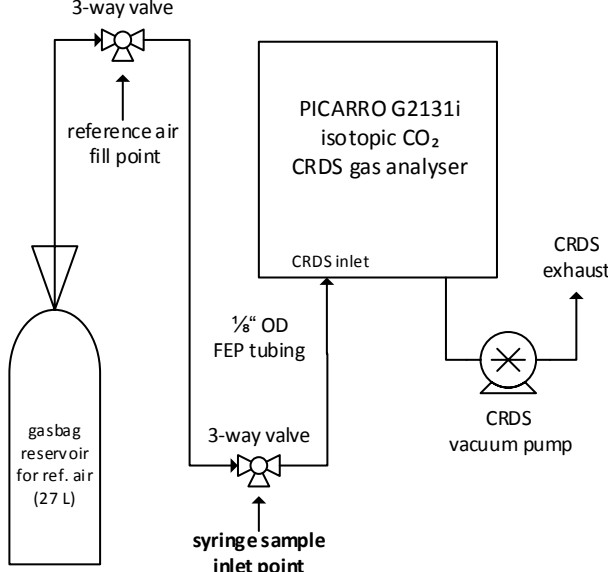

**Figure 1: Schematic diagram of the discrete gas sample measurement system coupled to the isotopic-CO₂ CRDS analyser.**



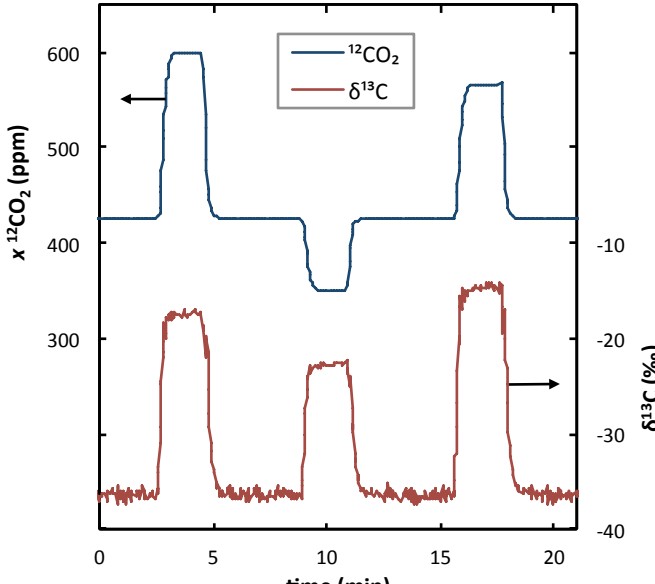

**Figure 2: Example CRDS data feed for syringe samples. Reference air measurements (ca. 425 ppm $x^{12}CO_2$ and -37 ‰ $\delta^{13}C$-$CO_2$) are interrupted by successive samples to form consistently identifiable peaks in the data.**





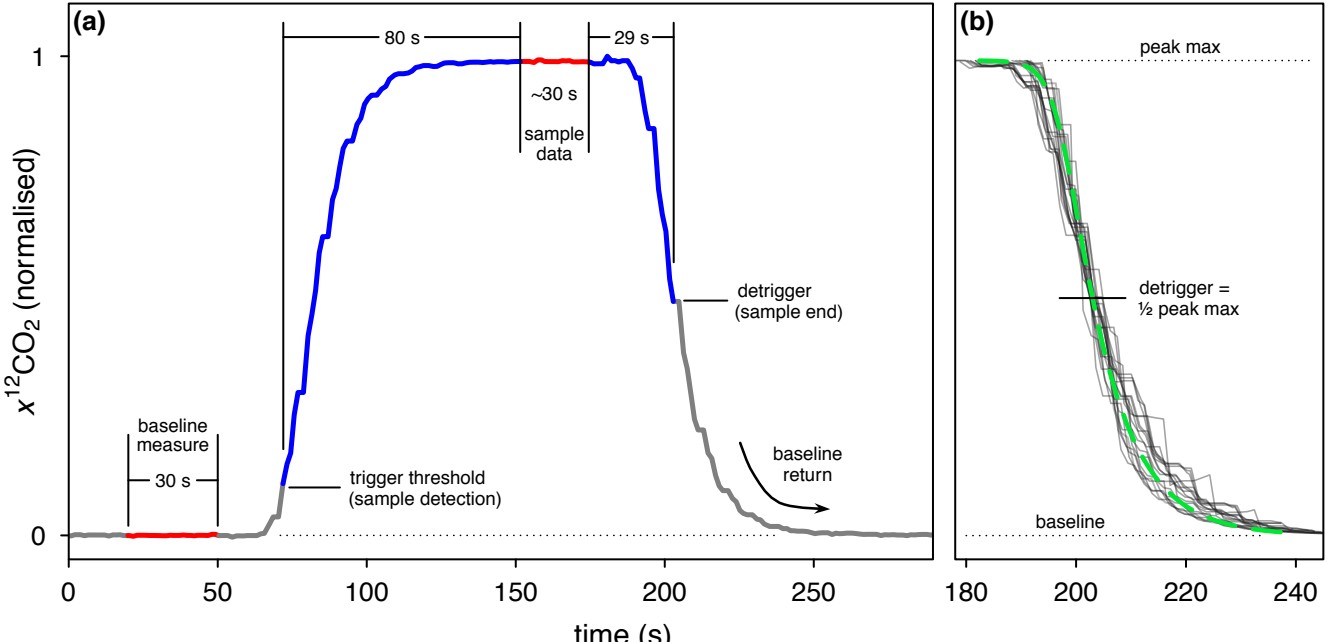

**Figure 3: (a) Example of raw G2131-i measurement data and breakdown of events during analysis of a 50 ml syringe sample. Blue segments are the data truncated from the sample peak and red the extracted measurements. All timings and thresholds are user-customisable in the measurement software for variation in sample size and equipment. (b) The most reliable sample end time (detrigger) was established as the point when measures returned to half the difference between peak-maximum (or minimum in the case of samples with lower $x$CO$_2$ than reference air) and the baseline value. Grey lines are amplitude-normalised tailing segments from 23 test samples varying in $x^{12}$CO$_2$. The broken green curve denotes a generalised logistic function fit to these test data by non-linear least squares optimisation. Solving the fitted function determined that 29 ± 2 s elapsed between peak-maximum and half-maximum irrespective of sample composition.**




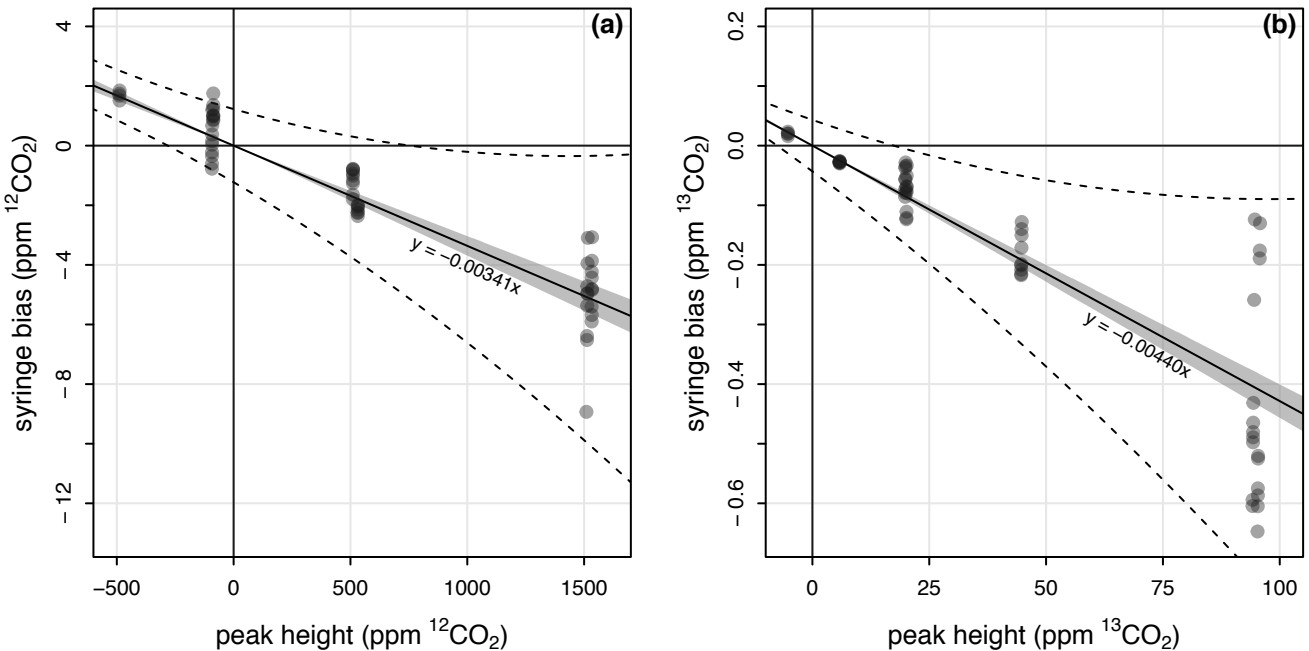

**Figure 4: Discrepancies between syringe sample and direct bottle measurements (syringe bias) of gas standards as a function of the syringe sample peak height (Eq. 7) for (a) $x^{12}CO_2$ and (b) $x^{13}CO_2$. The WLS fitted linear models (see Sect. 2.3) are overlaid for comparison (solid lines; slopes = 1-$K$, Eq. 7), with 95 % confidence intervals (shaded) and 95 % prediction intervals (dashed lines) as determined from the standard error estimates of $K_{C12}$ and $K_{C13}$ (Sect. 3.2).**



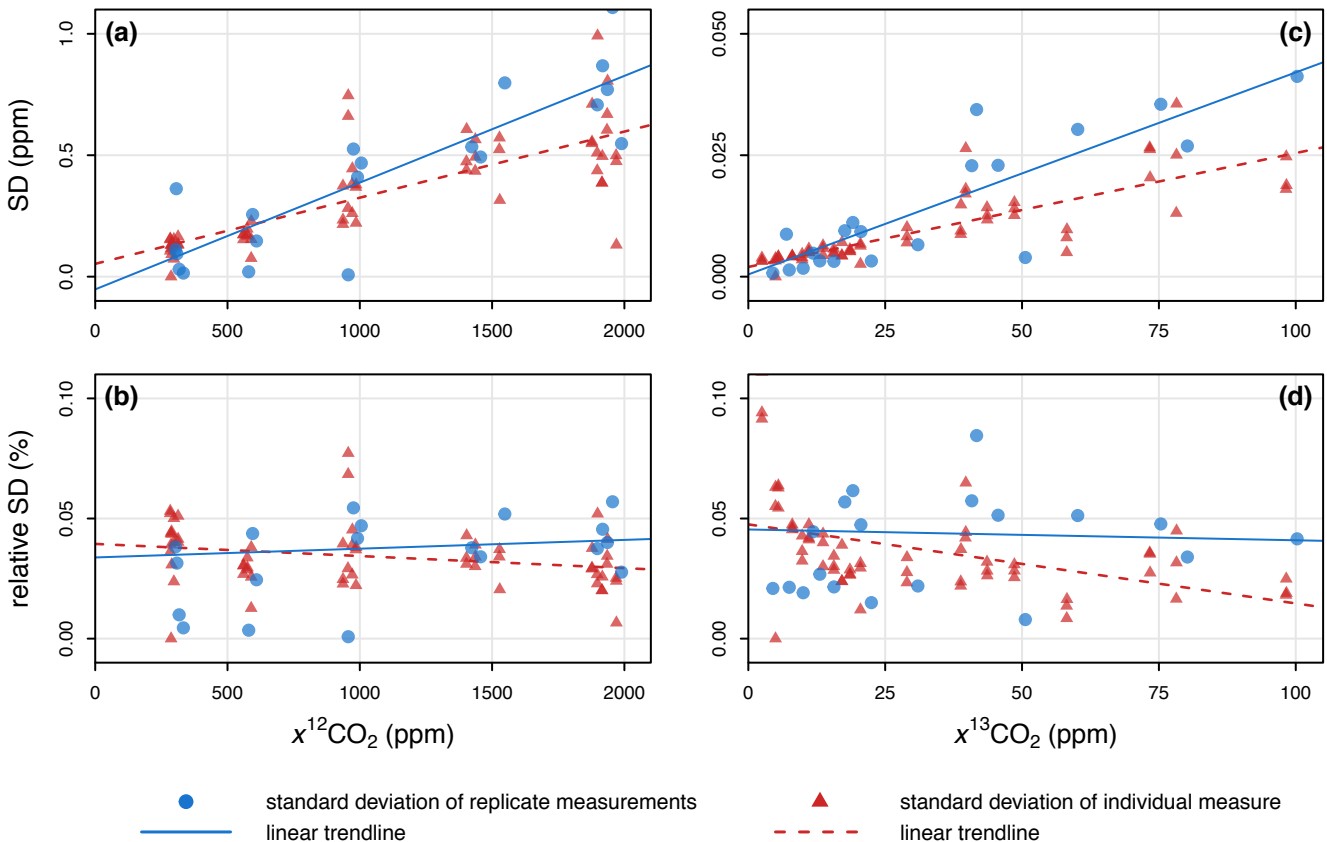

**Figure 5:** Precision in syringe sample data for $x^{12}CO_2$ (left: a, b) and $x^{13}CO_2$ (right: c, d) as quantified by standard deviations (top: a, c) and relative standard deviations (bottom: b, d) for individual measures (red) and replicate measurements (blue).





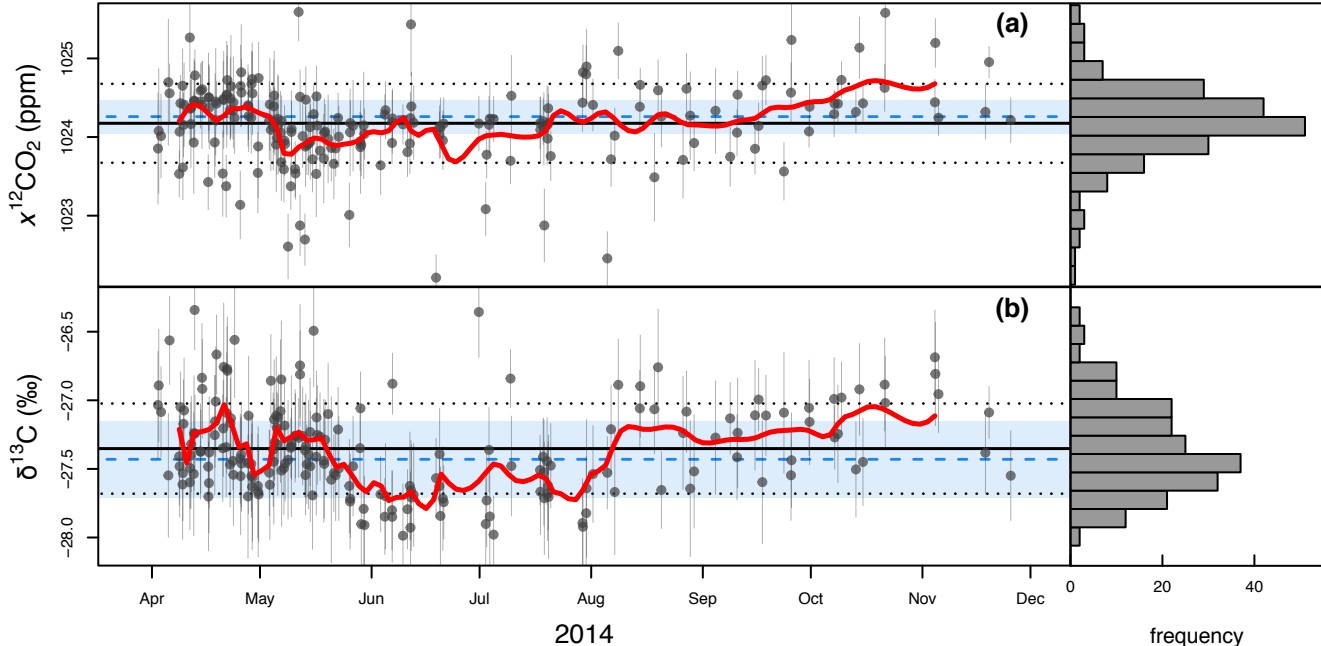

Figure 6: **Repeated syringe sample measurements in (a) $x^{12}CO_2$ and (b) $\delta^{13}C$-$CO_2$ of standard gas NA2 (Table 1) over a 9-month period (n = 200). Error bars denote ±1 SD of the data comprising each individual measure. Grand means are the solid black horizontal lines with dotted lines indicating ±1 SD of all measurements. Blue dashed lines and shaded areas indicate the direct bottle measurement values and ±1 SD. 10-measure moving averages are shown in red. Histograms inset right depict cumulative distributions of syringe measurements.**



| Standard ID | $x^{12}CO_2$ (ppm) | $x^{13}CO_2$ (ppm) | $xCO_2$ (ppm) | $R_{CO_2}$ ($^{13}CO_2/^{12}CO_2$)[*] | $\delta^{13}C\text{-}CO_2$ (‰)[**] |
|---|---|---|---|---|---|
| NA1 (Ref. air) | 490.55 (0.13) | 5.286 (0.004) | 495.84 (0.13) | 1.0776 (0.0006) | -36.14 (0.57) |
| NA2 | 1024.26 (0.21) | 11.137 (0.004) | 1035.39 (0.21) | 1.0874 (0.0003) | -27.43 (0.28) |
| ZERO | 0.05 (0.04) | 0.004 (0.004) | 0.05 (0.04) | - | - |
| HE1 | 2028.98 (0.47) | 25.528 (0.007) | 2054.51 (0.47) | 1.2582 (0.0004) | +125.35 (0.34) |
| HE2 | 2009.15 (0.53) | 100.11 (0.02) | 2109.26 (0.53) | 4.983 (0.001) | +3456.9 (1.1) |
| TT | 1002.18 (0.22) | 50.216 (0.008) | 1052.40 (0.22) | 5.011 (0.001) | +3481.7 (1.1) |
| LE1 | 402.24 (0.11) | 25.249 (0.005) | 427.49 (0.11) | 6.277 (0.002) | +4614.5 (1.7) |
| LE2 | 398.21 (0.16) | 101.24 (0.01) | 499.45 (0.16) | 25.42 (0.01) | +21739 (9) |

[*] $R_{CO_2}$ data are scaled by $10^2$ for ease of comprehension.

[**] $\delta^{13}C\text{-}CO_2$ values are reported against VPDB (Werner and Brand, 2001).

**Table 1: Bottle measurement data of the standard air used as baseline for syringe sample measures (NA1) and the gas standards used in method calibration (NA2 through LE2). Values are the averages (SDs in parentheses) of 10 min measurements taken for each standard directly inlet to the CRDS analyser (see Sect. 2.3). Data have been corrected as per the calibration of Dickinson et al. (2017).**