# Peer review of "System for $\delta^{13}$ C-CO2 and *x*CO2 analysis of discrete gas samples by cavity ring-down spectroscopy"

_Atmospheric Measurement Techniques, 2017_

## Referee Comment (RC1) · Anonymous Referee #1 · 17 May 2017

**Summary**

Excellent paper that improves upon a previous method to determine $^{13}$CO$_2$/$^{12}$CO$_2$ ratios and CO$_2$ concentration in discrete gas samples using a CRDS instrument (Picarro G2131-i). Methods are described in enough detail to replicate study and to apply to own research. Major advantages of this method over previous methods are: 1) high throughput time (though clarification needed for throughput time of previous method) and 2) increased precision of syringe samples (though clarification needed for high/variable ppm CO$_2$ applications). Disadvantages are larger sample sizes than past syringe methods (50 mL vs. 30 mL), higher time costs due to manual involvement of operator and determination of a specific system's K constants for correcting memory effects. However, I think the advantages will outweigh the disadvantages in many applications and this method should prove very helpful to future carbon cycle research.

[Figure]

This work is timely, important and novel, and I think only a few changes are needed for final publication.

**Highlights**

- Thorough consideration and measurement of memory effects, precision and consistency.

- Inclusion in supplementary info of worksheet for syringe bias correction.

- Nearly perfect writing that is easily understood and to-the-point.

**Areas for improvement**

- It isn't clear how samples of unknown different concentrations (such as those collected from headspace in a soil incubation experiment with different amounts of labile C, for example) can be accurately measured in the same run, because the authors state that the precision is best when the reference gas $CO_2$ mole fraction is similar to that of the sample gas and only differ in terms of $^{13}C$ content. It would have been helpful to have some discussion of this application in the paper, especially since the Introduction places this research into the context of headspace samples.

- Related to the first point, the reader would benefit from clarification regarding the reported $\delta^{13}C\text{-}CO_2$ precision. The precision seemed to be lowest with atmospheric samples (0.15‰), yet headspace samples are always higher in concentration than atmospheric. It would help if the authors were more clear about how the precision would increase with typical headspace concentration rates. On P10, L9-21, the authors report precisions of 0.33 to 0.35‰ for higher concentrations; I wonder if those would be more typical of what to expect with headspace samples.

**Line by line edits**

P2, L19: It would help place this in context if you could report the previous' systems sample turnover rates.

P3, L17: No capitalization needed for "Hermetic".

P3, L29: "insturment" should be corrected to "instrument".

P4, L15-L20: It might help to reader to explicitly state whether this data processing was done in real-time or after the run was completed. Specifically, I am wondering if the detrigger was detected in real-time, which would make manual operation of the instrument easier than if the user had to guess when the detrigger happened to know when to inject the next sample.

Figures 5 and 6: In the figure captions, I think that using the terms "intra-sample" and "inter-sample" would be better here and more obviously connect to the text in the results and discussion.

---

## Author Comment (AC1) · 24 May 2017

**Authors' reply to Anonymous Referee #1 interactive comment (RC1):**

Thank you for reviewing our manuscript. We appreciate both the encouraging comments and criticisms.

In terms of line-by-line edits, thank you for spotting the typos and areas of confusion. All of the points will be accommodated without difficulty in revision of the paper.

The more significant issues that were mentioned:

- 1. Confusion over the operation of the data-processing software script
- 2. Clarifying the application to soil headspace samples

**3. Ambiguity over the achieved precision of the method**

Regarding point 1: Our software script operates in real-time – processing data, monitoring the measurement process, and prompting the user to introduce syringe samples at the correct time. We will re-write part of the methods section and emphasise the utility of the script. We also intend to produce a short demonstration video to accompany the paper, which should provide the audience with a clear understanding of the measurement procedure and software.

For point 2: We will improve the discussion to explicitly mention application to headspace samples and likely performance in such cases.

For point 3: There has been a misunderstanding of method performance / precision. This was not helped by the unintended omission of the caption to Supplementary Figure S1. We will revise our discussion and other relevant sections to clarify our findings and reduce the risk of confusion in the final paper. For completeness, here below we answer specific issues that were queried on this point:

- Measurement precision is independent of reference gas composition. There is no
  effect on precision (or accuracy) by measuring a sequence of samples with CO2
  conc. very variable and different to the reference gas compared to a sequence
  of uniform samples similar to the reference gas. We meant to report that the
  method works best in terms of sample throughput rate when CO2 concentration
  of the sample is similar to the reference gas, not that measurement precision was
  improved.
- For headspace samples much higher in CO2 conc. than the reference gas, the only real challenge is a potential slow-down in the sample throughput rate due to increased inter-sample waiting time for memory effects dissipate (NB: this is
a separate issue post-correction of memory effects in actual sample measurements). Our software script monitors the CRDS data-stream, in real-time, and ensures that memory effects from the previous sample are gone before prompting the operator to introduce the next sample. The bigger the difference in  $CO_2$ composition between the reference and the sample, the longer the waiting time between samples (thus reducing sample turnover rate). For instance, if the reference gas is 500 ppm CO2 and samples ranged 400-600 ppm (all natural  $^{13}$ C abundances), measuring 12 samples  $h^{-1}$  is realistic. However, if the difference in  $CO_2$  conc. between samples and the reference were larger, e.g. if the reference is 500 ppm and samples 2000-3000 ppm, then throughput would reduce to around 8 samples  $h^{-1}$ . With a sample CO2 conc. >6000 ppm, the memory effect after each sample takes perhaps 15 minutes to clear, and so throughput would be <5 samples h-1. The situation is similar for samples highly enriched in 13C. An additional (but separate) issue in such examples is that because the normal operating range of G2131-i/G2201-i is 380-2000 ppm and natural abundance 13 C, actual CRDS accuracy may become a question at very high concentrations or  $^{13}$ C-enrichments (although we found that up to ca. 5000 ppm and +2000 ‰ vs. VPDB, CRDS accuracy is still very good).

• In terms of precision, the repeatability for CO2 concentration measurements is ca. 0.05 % of the measured value, irrespective of the actual concentration (e.g. it's 0.2 ppm for 400 ppm samples, 1 ppm at 2000 ppm). An exception is at low concentrations (e.g. CO2 <100 ppm) when CRDS resolution holds constant in absolute terms at around 0.02-0.05 ppm instead of a relative 0.05 %. Precision in  $\delta^{13}$ C is difficult to communicate because  $\delta^{13}$ C is a relative measure itself and describing the precision of a relative measure becomes confusing and misleading when a large range of values is covered, as in our precision tests. The most important point to understand is that, all else being equal, higher CO2 conc. improves precision in isotope ratio measurement. For samples with natural 13C
abundance and atmospheric CO2 conc., we found that precision in  $\delta^{13}$ C is ca. 0.15 ‰ (inter-sample SD). The reported 0.33 ‰ is the SD of 200 samples over a 9-month period. However, that is not a good indication of the repeatability from successive samples during 1 day in the lab. The gap between 0.15 ‰ and 0.33 ‰ shows the additional presence of small random instrument/methodological drift (inaccuracy) over the course of 9-months of operation. The report of 0.35 ‰ is the mean intra-sample SD of the 200 samples, and this value does indeed match our observations from the systematic precision testing. For the case of soil headspace samples (with natural abundance 13C-CO2), if multiple samples are taken at a time, then precision of ca. 0.15 ‰ can be expected (inter-sample SD). However if only one headspace sample SD of that single sample, which will be ca. 0.3 ‰ . At higher CO2 concentrations, both these precisions may improve slightly.

---

## Referee Comment (RC2) · Anonymous Referee #2 · 26 Jun 2017

Dickinson et al. present a new and rather simple method that can easily analyze small discrete gas samples using a commercially available cavity ring-down spectroscopy gas analyzer. The major advancement in the performance of the system, compared to other methods, is a two-fold improvement in the throughput rate, which may be appreciated when such a system is regularly used for analysis of a large number of samples in the laboratory, as is the case described in the manuscript. Although it was developed for analysis of xCO2 and $\delta$13C-CO2, the method can be extended to analyze other species with similar instrumentation. My general impression is that the real content of the manuscript is thin, and a significant part of the text focuses on apparent technical description/maintenance rather than technical advancement. For example, it is unclear whether there is any advantage in the precision/accuracy of

the system compared to other methods, other than the precision improvement of the commercially available CRDS itself. The accuracy of the measurements is not included due to the separation of one story into two manuscripts that are simultaneously in review for two different journals, which I found it, at several places, inconvenient to be forced to read another manuscript of the same author to obtain necessary details. Considering the abovementioned points, I strongly recommend (even I know it is hard to convince) the authors combining the two manuscripts and publish one piece of nice work. One good paper is worth more than two OK papers.

Other comments:

1) Comparing the precision of the system and that of previous systems, how much of the improvement is due to the enhanced spectroscopic sensitivity of the CRDS?

2) The method uses ∼30 sample data for the analysis. Have the authors considered making a curve fit to the data set and using the steady value of the fit instead? In this way, the measurement will not be sensitive to the baseline signal any more.

Detailed comments:

P3/L29: what does "stable operation" imply here? As the cavity temperature is strictly controlled, is any difference expected if the whole system is located in an unconditional room?

P5/L26: Can the authors explain why zero air (0.05 ppm $CO_2$) is included and why is the range claimed to include the zero air? I do not see the value of adding zero air, and the isotopic signature of the zero seems strange.

P5: I wonder whether there is systematic but significant bias between the "true" value of the syringe sample and the bottle sample, which could be introduced during the sampling process.

P10/L10: Were the 9-month period measurements calibrated? It is difficult to judge when the accuracy of the system is not mentioned in the manuscript.

P10/L31: The traditional continuous-flow IRMS can do much better than ∼0.1‰. The reference should not be limited to an old paper Prosser et al., 1991.

---

## Author Comment (AC3) · 1 Sep 2017

**Final author comments to peer review and manuscript revisions**

We would like to once again thank the two anonymous referees and associate editor for reviewing our work and providing both encouraging and critical comments.

We have directly responded to each of the referee comments with author replies. From these and editor comments, we have subsequently revised our paper. Here below we list the relevant amendments and improvements. Supplement to this we have uploaded a copy of the revised manuscript with all changes tracked and marked.

**Significant changes in revised manuscript:**

- Clarified all discussion points about achieved precision of our method and eliminated the confusion stemming from comments about sample-throughput performance (sample turnover rate and measurement precision are entirely independent of each other).

- Added a new discussion section (Sect. 3.4) to outline the potential applications of the method (with particular regard to soil headspace studies). Made clear expected precision for sample measurements in headspace studies:

  - For natural abundance samples, precision in $\delta^{13}$C-CO$_2$ of repeated measures (inter-sample) is ca. 0.15 ‰ .

  - The precision of a single sample measurement (intra-sample) will be ca. 0.2 to 0.5 ‰ (which reflects the noise of the CRDS analysis over the short 30s measurement period of the sample) and this also depends upon $x$CO$_2$ level (higher $x$CO$_2$ gives better precision; Fig. S1a).

- Have better explained the functionality of our software script that manages the measurement process:

  - The script's data analysis works in real-time.

  - The trigger and detrigger points are detected and the operator prompted accordingly.

  - The software directs the user when to introduce the samples.

**Minor changes in revised manuscript:**

- Incorporated comparisons of previous methods' sample throughputs in order to give context to performance of our system.

- Altered legends for Figs. 5, 6, and S1 to accord the "inter-sample" and "intra-sample" terminology used in the body text for describing measurement precision / data variance. We also have rectified the unintended omission of the caption to Fig. S1.

- We have adjusted our reports of precision so as to not give the impression that our measurement method provides better precision than previous / other methods. We have more explicitly explained that precision achieved in our method chiefly reflects the precision of the underlying CRDS analyser.

- Clarified and more thoroughly discussed the sources of variance in the long-term repeated measurements data (9-month period, 200 measures). More directly explained that the observed increase in variance in these data is likely due to instrument drift but could equally be due to transient inconsistencies in the syringe method.

- Removed the reference of CF-IRMS measurement performance (Prosser et al., 1991) to avoid a direct performance comparison.

- Various improvements in wording, grammar. Fixes of typos and formatting mistakes.

**Other comments:**

- One point raised during peer-review concerned our citation of a separate paper of ours that we recently submitted to *Rapid Communications in Mass Spectrometry* (RCM). That paper covered calibration of CRDS gas analyser under conditions of highly enriched $^{13}$C abundance. We are pleased to report that the peer-review process of that paper is completed and publication in RCM is forthcoming. http://onlinelibrary.wiley.com/doi/10.1002/rcm.7969/full

- We have produced a short supplementary video showing our method in operation. This provides a demonstration of how to perform the syringe measurements as well as visual depiction of the physical measurement set-up. The video is currently available at (later to become formal video supplement): https://youtu.be/jqVFUO-EuCk

Please also note the supplement to this comment:
https://www.atmos-meas-tech-discuss.net/amt-2017-57/amt-2017-57-AC3-supplement.pdf

**Supplement:**

[revised manuscript text omitted]

20 *Acknowledgements.*

We are grateful to our ISOFYS colleagues Stijn Vandevoorde, Hannes De Schepper, and Katja Van Nieuland for their assistance with instrument operation and numerous sample measurements. We also thank Lei Liu (CREAF-CSIC, Barcelona, Spain) for testing our method on a Picarro G2201-i CRDS unit and for providing feedback on method efficacy. Renato Winkler from Picarro Inc. aided this work with his useful advice on developing software scripts for the G2131-i analyser.
25 Finally, we thank two anonymous referees whose comments helped refine this paper.

**References**

Albanito, F., McAllister, J. L., Cescatti, A., Smith, P., and Robinson, D.: Dual-chamber measurements of $\delta^{13}C$ of soil-respired $CO_2$ partitioned using a field-based three end-member model, Soil Biol. Biochem., 47, 106-115, 2012.

Dane Dickinson 29/8/2017 21:25

Dane Dickinson 29/8/2017 21:25

Dane Dickinson 29/8/2017 21:25

Dane Dickinson 29/8/2017 21:25

Dane Dickinson 29/8/2017 21:25

Dane Dickinson 29/8/2017 21:25

Dane Dickinson 29/8/2017 21:25

Dane Dickinson 29/8/2017 21:25

Dane Dickinson 29/8/2017 21:25

Dane Dickinson 29/8/2017 21:25

[revised manuscript text omitted]

---

## Author Comment (AC4) · 7 Sep 2017

We have produced a short video showing our method in operation. This provides a demonstration of how to perform the syringe measurements as well as visual depiction of the physical measurement set-up.

The video is currently available at:

https://youtu.be/jqVFUO-EuCk

---

## Author Response (AR1)

**Authors' response to to peer review**

We would like to once again thank the two anonymous referees and associate editor for reviewing our work and providing both encouraging and critical comments.

We have directly responded to each of the referee comments with author replies. We have also already replied directly to the editor on related matters. Below we discuss a specific issue concerning suitability for publication in AMT.

From all this, we have subsequently revised our paper. Further below we list the relevant amendments and improvements.

After the change lists, we attach a copy of the revised manuscript with all changes tracked and marked.

We also attach copies of our earlier replies to reviewers.

**Specific note to editor:**

One of the referees suggested that we should combine two papers into one (the editor also made a similar intimation).

Our response to this notion:

A combined publication of our work would be inappropriate for 4 key reasons:
- First, our other paper in RCM concerns only measurements of $^{13}$C-enriched $CO_2$ by CRDS. This is not a subject relevant to AMT and is primarily interesting only to researchers working with $^{13}$C-tracers. That the present paper here draws on that other work is unimportant for a researcher wishing to make syringe sample measurements of natural isotopic abundance $CO_2$. The present paper could be written to exclude any mention of our other work although we see no good reason to do this. The purpose of using $^{13}$C-enriched standards in this work was to test syringe measures over a wide range of $CO_2$ compositions and to investigate the possibility of different isotopolouges having different (independent) behaviour during infilling of the optical cavity (which would lead to additional error in isotope ratio measurements of small samples).
- Second, the two manuscripts address entirely separate causes of inaccuracy in gas measurements that should be corrected separately. The present work primarily documents a new method for making discrete sample measurements by continuous sampling CRDS instruments and thus the key inaccuracy to address is contamination / memory effects from the syringe method. The other paper only covers errors from spectroscopic cross-talk between $^{12}CO_2$ and $^{13}CO_2$ isotopologues.
- Third, the body of text for such a combined paper (excluding captions and references) would comprise in excess of 8000 words, as it would describe two unrelated physical processes with very little overlap. In addition, there would be several dozen supplementary files. We do not share the reviewer's view that one

long cumbersome paper would be more valuable than two shortish papers each addressing separate and compartmentalised problems.
- Fourth but most importantly, our other publication is accepted in RCM and is forthcoming:
    http://onlinelibrary.wiley.com/doi/10.1002/rcm.7969/full

**Significant changes in revised manuscript:**

- Clarified all discussion points about achieved precision of our method and eliminated the confusion stemming from comments about sample-throughput performance (sample turnover rate and measurement precision are entirely independent of each other).
- Added a new discussion section (Sect. 3.4) to outline the potential applications of the method (with particular regard to soil headspace studies). Made clear expected precision for sample measurements in headspace studies:
    o For natural abundance samples, precision in $\delta^{13}$C-CO$_2$ of repeated measures (inter-sample) is ca. $0.15\,‰$ .
    o The precision of a single sample measurement (intra-sample) will be ca. 0.2 to 0.5 ‰ (which reflects the noise of the CRDS analysis over the short 30s measurement period of the sample) and this also depends upon $x$CO$_2$ level (higher $x$CO$_2$ gives better precision; Fig. S1a).
- Have better explained the functionality of our software script that manages the measurement process:
    o The script's data analysis works in real-time.
    o The trigger and detrigger points are detected and the operator prompted accordingly.
    o The software directs the user when to introduce the samples.

**Minor changes in revised manuscript:**

- Incorporated comparisons of previous methods' sample throughputs in order to give context to performance of our system.
- Altered legends for Figs. 5, 6, and S1 to accord the "inter-sample" and "intra-sample" terminology used in the body text for describing measurement precision / data variance. We also have rectified the unintended omission of the caption to Fig. S1.
- We have adjusted our reports of precision so as to not give the impression that our measurement method provides better precision than previous / other methods. We have more explicitly explained that precision achieved in our method chiefly reflects the precision of the underlying CRDS analyser.
- Clarified and more thoroughly discussed the sources of variance in the long-term repeated measurements data (9-month period, 200 measures). More directly explained that the observed increase in variance in these data is likely due to instrument drift but could equally be due to transient inconsistencies in the

syringe method.
- Removed the reference of CF-IRMS measurement performance (Prosser et al., 1991) to avoid a direct performance comparison.
- Various improvements in wording, grammar. Fixes of typos and formatting mistakes.

**Other comments:**

- One point raised during peer-review concerned our citation of a separate paper of ours that we recently submitted to *Rapid Communications in Mass Spectrometry* (RCM). That paper covered calibration of CRDS gas analyser under conditions of highly enriched $^{13}C$ abundance. We are pleased to report that the peer-review process of that paper is completed and publication in RCM is forthcoming.
  http://onlinelibrary.wiley.com/doi/10.1002/rcm.7969/full
- We have produced a short supplementary video showing our method in operation. This provides a demonstration of how to perform the syringe measurements as well as visual depiction of the physical measurement set-up. The video is currently available at (later to become formal video supplement):
  https://youtu.be/jqVFUO-EuCk

[revised manuscript text omitted]

20 *Acknowledgements.*

We are grateful to our ISOFYS colleagues Stijn Vandevoorde, Hannes De Schepper, and Katja Van Nieuland for their assistance with instrument operation and numerous sample measurements. We also thank Lei Liu (CREAF-CSIC, Barcelona, Spain) for testing our method on a Picarro G2201-i CRDS unit and for providing feedback on method efficacy. Renato Winkler from Picarro Inc. aided this work with his useful advice on developing software scripts for the G2131-i analyser.
25 Finally, we thank two anonymous referees whose comments helped refine this paper.

**References**

Albanito, F., McAllister, J. L., Cescatti, A., Smith, P., and Robinson, D.: Dual-chamber measurements of $\delta^{13}C$ of soil-respired $CO_2$ partitioned using a field-based three end-member model, Soil Biol. Biochem., 47, 106-115, 2012.

Dane Dickinson 29/8/2017 21:25

Dane Dickinson 29/8/2017 21:25

Dane Dickinson 29/8/2017 21:25

Dane Dickinson 29/8/2017 21:25

Dane Dickinson 29/8/2017 21:25

Dane Dickinson 29/8/2017 21:25

Dane Dickinson 29/8/2017 21:25

Dane Dickinson 29/8/2017 21:25

Dane Dickinson 29/8/2017 21:25

Dane Dickinson 29/8/2017 21:25

[revised manuscript text omitted]

**Authors' reply to Anonymous Referee #1 interactive comment (RC1):**

Thank you for reviewing our manuscript. We appreciate both the encouraging comments and criticisms.

In terms of line-by-line edits, thank you for spotting the typos and areas of confusion. All of the points will be accommodated without difficulty in revision of the paper.

The more significant issues that were mentioned:
    (a) Confusion over the operation of the data-processing software script
    (b) Clarifying the application to soil headspace samples
    (c) Ambiguity over the achieved precision of the method

Regarding point (a): Our software script operates in real-time – processing data, monitoring the measurement process, and prompting the user to introduce syringe samples at the correct time. We will re-write part of the methods section and emphasise the utility of the script. We also intend to produce a short demonstration video to accompany the paper, which should provide the audience with a clear understanding of the measurement procedure and software.

For point (b): We will improve the discussion to explicitly mention application to headspace samples and likely performance in such cases.

For point (c): There has been a misunderstanding of method performance / precision. This was not helped by the unintended omission of the caption to Supplementary Figure S1. We will revise our discussion and other relevant sections to clarify our findings and reduce the risk of confusion in the final paper. For completeness, here below we answer specific issues that were queried on this point:

- Measurement precision is independent of reference gas composition. There is no effect on precision (or accuracy) by measuring a sequence of samples with $CO_2$ conc. very variable and different to the reference gas compared to a sequence of uniform samples similar to the reference gas. We meant to report that the method works best in terms of sample throughput rate when $CO_2$ concentration of the sample is similar to the reference gas, not that measurement precision was improved.

- For headspace samples much higher in $CO_2$ conc. than the reference gas, the only real challenge is a potential slow-down in the sample throughput rate due to increased inter-sample waiting time for memory effects dissipate (NB: this is a separate issue post-correction of memory effects in actual sample measurements). Our software script monitors the CRDS data-stream, in real-time, and ensures that memory effects from the previous sample are gone before prompting the operator to introduce the next sample. The bigger the difference in $CO_2$ composition between the reference and the sample, the longer the waiting time between samples (thus reducing sample turnover rate). For instance, if the reference gas is

500 ppm $CO_2$ and samples ranged 400-600 ppm (all natural $^{13}C$ abundances), measuring 12 samples $h^{-1}$ is realistic. However, if the difference in $CO_2$ conc. between samples and the reference were larger, e.g. if the reference is 500 ppm and samples 2000-3000 ppm, then throughput would reduce to around 8 samples $h^{-1}$. With a sample $CO_2$ conc. >6000 ppm, the memory effect after each sample takes perhaps 15 minutes to clear, and so throughput would be <5 samples $h^{-1}$. The situation is similar for samples highly enriched in $^{13}C$. An additional (but separate) issue in such examples is that because the normal operating range of G2131-i/G2201-i is 380-2000 ppm and natural abundance $^{13}C$, actual CRDS accuracy may become a question at very high concentrations or $^{13}C$-enrichments (although we found that up to ca. 5000 ppm and +2000 ‰ vs. VPDB, CRDS accuracy is still very good).

- In terms of precision, the repeatability for $CO_2$ concentration measurements is ca. 0.05 % of the measured value, irrespective of the actual concentration (e.g. it's 0.2 ppm for 400 ppm samples, 1 ppm at 2000 ppm). An exception is at low concentrations (e.g. $CO_2$ <100 ppm) when CRDS resolution holds constant in absolute terms at around 0.02-0.05 ppm instead of a relative 0.05 %. Precision in $\delta^{13}C$ is difficult to communicate because $\delta^{13}C$ is a relative measure itself and describing the precision of a relative measure becomes confusing and misleading when a large range of values is covered, as in our precision tests. The most important point to understand is that, all else being equal, higher $CO_2$ conc. improves precision in isotope ratio measurement. For samples with natural $^{13}C$ abundance and atmospheric $CO_2$ conc., we found that precision in $\delta^{13}C$ is ca. 0.15 ‰ (inter-sample SD). The reported 0.33 ‰ is the SD of 200 samples over a 9-month period. However, that is not a good indication of the repeatability from successive samples during 1 day in the lab. The gap between 0.15 ‰ and 0.33 ‰ shows the additional presence of small random instrument/methodological drift (inaccuracy) over the course of 9-months of operation. The report of 0.35 ‰ is the mean intra-sample SD of the 200 samples, and this value does indeed match our observations from the systematic precision testing. For the case of soil headspace samples (with natural abundance $^{13}C$-$CO_2$), if multiple samples are taken at a time, then precision of ca. 0.15 ‰ can be expected (inter-sample SD). However if only one headspace sample is taken at each time-point, the only precision value available is the intra-sample SD of that single sample, which will be ca. 0.3 ‰ . At higher $CO_2$ concentrations, both these precisions may improve slightly.

**Authors' reply to Anonymous Referee #2 interactive comment (RC2):**

NB: Original referee comments in black text. Author comments in red text.

We thank the reviewer for critiquing/commenting on our manuscript. We have included the complete text of RC2 below and made embedded replies in red so as to address directly the comments in context.

Dickinson et al. present a new and rather simple method that can easily analyze small discrete gas samples using a commercially available cavity ring-down spectroscopy gas analyzer. The major advancement in the performance of the system, compared to other methods, is a two-fold improvement in the throughput rate, which may be appreciated when such a system is regularly used for analysis of a large number of samples in the laboratory, as is the case described in the manuscript. Although it was developed for analysis of xCO2 and δ13C-CO2, the method can be extended to analyze other species with similar instrumentation. My general impression is that the real content of the manuscript is thin, and a significant part of the text focuses on apparent technical description/maintenance rather than technical advancement. For example, it is unclear whether there is any advantage in the precision/accuracy of the system compared to other methods, other than the precision improvement of the commercially available CRDS itself. The accuracy of the measurements is not included due to the separation of one story into two manuscripts that are simultaneously in review for two different journals, which I found it, at several places, inconvenient to be forced to read another manuscript of the same author to obtain necessary details. Considering the abovementioned points, I strongly recommend (even I know it is hard to convince) the authors combining the two manuscripts and publish one piece of nice work. One good paper is worth more than two OK papers.

We understand the impression of the reviewer – it might seem like a trivial adaptation to transform a continuous flow instrument into a discrete analyser. However, we strongly believe that there is considerable need for a detailed description of 'simple' discrete sample laser based isotope analyser. At present, there is no time and cost effective method for reliably measuring discrete gas samples by continuous sampling CRDS instruments such as the Picarro G2131-i and other models. Commercial peripheries (e.g. Picarro A0314 SSIM2) and previous published method (Berryman et al. 2011) are slow, complex, and cannot provide gas mole fraction data due to dilution processes inherent to the measurement process. There is clear need (in soil respiration headspace studies as just

one example) for a practical simple way to make accurate measurements of small discrete samples with CRDS instruments (both for isotope ratio and mole fraction measurements).

The rationale given for the reviewer's concerns were was as follows:
1. That the work does not constitute a technical advancement.
2. That the paper does not properly compare the precision and accuracy of the presented method with previous/other methods.
3. That accuracy of our system is not addressed, which is instead referred to in a separate publication.

To point 1:
To the best of our knowledge there is no published description of an equivalent method for conducting discrete gas sample measurements by CRDS instruments at a rate of 12 $h^{-1}$ that gives both accurate isotope ratio and mole fraction data. Hence we stand by our work as an important advance to the state-of-the-art.

To point 2:
This is not correct – we have compared our method against existing methods and equipment (Sect. 3.3). We do not make major claim that our method significantly improves precision compared to other methods, but we do report the precision we achieved, and we note that it is at least similar to other methods. As for accuracy, any measurement system or method that is "properly calibrated" is "accurate", by definition. In addition to performing an appropriate calibration, we have reported the uncertainty associated with applying our calibration to correct for memory effects inherent in syringe measures (p. 9: 0.02% of the sample peak height).

And to point 3:
This is not correct – we have addressed the accuracy / bias of our method in Sections 2.3, 3.2, and Figure 4. It is true however that we have not addressed the accuracy of CRDS instrumentation in this work. We believe that such a question should be examined separately so as to not confuse or conflate the multiple phenomena that may cause errors in different CRDS measurements. There are numerous published papers that evaluate accuracy / calibration of CRDS instruments. Researchers that do not need to measure $CO_2$ compositions with high $^{13}C$ abundances will not find our other publication interesting, but they may nonetheless wish to perform measurements of small discrete syringe samples and find the present work extremely useful. A vice-versa scenario is also probable.

Other comments:

1) Comparing the precision of the system and that of previous systems, how much of the improvement is due to the enhanced spectroscopic sensitivity of the CRDS?

We do not know. We have explicitly acknowledged that the improved precision we report

may be due to improved CRDS instrumentation rather than advantage in our method. We do not mean to claim that our method gives significant advancement in precision (but it is important that our method is not worse in precision). Our primary claims are: high throughput rate, accurate simultaneous mole fraction and isotope ratio data, practicality, low cost, time-efficiency. In revision we will adjust some of the text to make our reports of achieved precision more modest.

2) The method uses ~30 sample data for the analysis. Have the authors considered making a curve fit to the data set and using the steady value of the fit instead? In this way, the measurement will not be sensitive to the baseline signal any more.

We understand this suggestion to mean that a steady baseline reading might not be necessary if we used a curve-fitting algorithm on the syringe sample data. We did think about this, but we foresaw two major problems:
- First, gas replacement / mixing in the optical cavity entails that the composition in the cavity prior to introduction of a syringe sample affects the CRDS measurement (memory effect), and consequently, for such an algorithm to work, the CRDS data prior to the syringe sample introduction would need to be an input variable. This is practically the same as recording the baseline.
- Secondly, designing a software script to perform such a task in real-time is not trivial. Aside from such curve-fitting probably requiring computationally expensive non-linear optimisation, Picarro instrumentation and software is not user-friendly for real-time data flagging and analysis. Yet in order to realise the suggestion, the fitting algorithm would need to "know" the exact time when the syringe sample was introduced into the analyser so as to provide a start-point. Building a computerised device to signal the position of the manual syringe input valve is not a simple solution in comparison to our baseline recording and peak detection process.

Detailed comments:

P3/L29: what does "stable operation" imply here? As the cavity temperature is strictly controlled, is any difference expected if the whole system is located in an unconditional room?

It is true that the optical cavity is well controlled, however other researchers have nevertheless noted environmentally induced variations in measurements, which are thought to arise out of residual uncompensated fluctuations to the cavity (Kwok et al. 2015). Ambient temperature fluctuation is also mentioned as a potential source of instrument drift in pamphlets published by the instrument manufacturer. An environmentally controlled lab simply mitigates all risk for error in this regard.

P5/L26: Can the authors explain why zero air (0.05 ppm CO2) is

included and why is the range claimed to include the zero air? I do not see the value of adding zero air, and the isotopic signature of the zero seems strange.

The greater the range of data used in the WLS optimisation of Eqs. 2-5, the lower the resulting uncertainty for correcting syringe bias / memory effects. By measuring zero air, we acquired excellent "negative peak" data, which thus improved the statistical estimates of the correction constants $K_{C12}$ and $K_{C13}$ (see Fig. 4). In terms of isotope ratio signature for zero air, well there is no sensible/measurable ratio that can be made: isotopic ratio "measurements" of zero air must be recognised as spurious given the CRDS instrument develops too much noise at ppb levels of $^{13}CO_2$ for meaningful ratio assessments. (Isotope ratio data for ZERO were excluded from WLS optimisation.)

P5: I wonder whether there is systematic but significant bias between the "true" value of the syringe sample and the bottle sample, which could be introduced during the sampling process.

We compared the syringe sample values against CRDS measurements of bottle standards (not against gravimetric values of the standards). The calibration/post-correction therefore transforms "syringe measurements" into "bottle measurements" eliminating the systematic bias between those two gas delivery methods. Any constant bias introduced by the syringe sampling process (e.g. ambient air contamination) would be seen as a liner offset (constant term) within the dataset shown in Figure 4, however no such offset was observed. Any other error or "inconsistent bias" from sampling would simply add to the random errors of the syringe measurements (and give worse inter-sample precision).

P10/L10: Were the 9-month period measurements calibrated? It is difficult to judge when the accuracy of the system is not mentioned in the manuscript.

Each individual sample measurement from the 9-month dataset was calibrated for syringe bias, but was not individually calibrated for random instrument drift. The reviewer is correct in noting that these data are therefore a simultaneous test of method accuracy and instrument accuracy. However, the purpose of these data is to examine consistency of the syringe method under typical laboratory practices over a long period of time. We have explicitly explained that the observed increase in variance seen in these data is likely due to instrument drift but could equally be due to transient inconsistencies in the syringe method. We will further clarify this point in revision.

P10/L31: The traditional continuous-flow IRMS can do much better than ~0.1‰The reference should not be limited to an old paper Prosser et al., 1991.

The reference was simply mentioned as a guideline value: From our experience and with

current information of CF-IRMS producers, 0.1‰ is a typical value and not entirely obsolete. However, to avoid any misrepresentation, we will remove this out-dated reference and avoid making a direct performance comparison to state-of-the-art CF-IRMS.

---

## Author Response (AR2)

**Authors' response to to editor decision**

**Associate Editor Decision: Publish subject to minor revisions (Editor review)** (26 Sep 2017) by David Griffith
Comments to the Author:
I thank the authors for providing an advanced copy of the preceding paper in RCMS on which the CRDS measurements are based. Having now seen this paper, I agree that the two papers can be separated as they address different aspects of the measurement problem. However, taking referee #2's comments into account, I suggest that a succinct summary of the RCMS paper be provided in the methods section, so that the reader can learn all that is needed to interpret the present paper without need to read the RCMS paer. This need only be a short paragraph. It should clearly set out the precision and accuracy of the CRDS measurement which underlies the present paper. In this way the extensive discussion of precision and accuracy of the syringe sampling method can be clearly distinguished from the underlying analyser precision and accuracy.

Thank you for the positive decision concerning our paper -- we believe the work is a valuable contribution to AMT and is now up to standard for publication.We have revised the manuscript to directly address the matter outlined above regarding on our paper in RCM. (See tracked changes below.)

One remaining outstanding item concerns including /excluding the supplementary video as a formal attachment (?)

**Latest changes in manuscript:**

Section **2.3 Measurement calibration**
- Added explanatory paragraph at top of section to give a short introduction into the various errors that CRDS measurements experience.
- Further clarified the reasoning behind and process for calibrating the memory effect bias.
- Gave short description of main findings from our RCM paper (Dickinson et al. 2017) and its relevance to the current work.

Section **2.4 Precision and consistency tests**
- Provided explanatory paragraph of the concept of precision with respect to CRDS and this work (moved and revised previous paragraph from Section 3.3).

Section **3.3 Measurement precision and consistency**
- Deleted leading explanatory paragraph (moved to section 2.4) for overall improved flow.
- Further clarified the fact that precision in our results derive from both the stability of the underlying CRDS analyser and the syringe sampling method.
- Directly warned the reader to beware of extrapolating our results to replica systems using a different CRDS analyser as the analyser is a significant portion of achieved precision.
- Provided guideline values for expected CRDS precision under "normal" usage so as to make a point of reference for the results presented in this work.

[revised manuscript text omitted]

Conventional CRDS trace gas measurements are affected by signal noise, gradual instrument drift, and any unaddressed
20  interferences or perturbations in the underlying spectroscopy (Vogel et al., 2013). Calibration strategies exist to counter both drift and spectral errors while random noise ultimately limits instrument precision (Friedrichs et al., 2010; Wen et al., 2013). In the present case of short discrete sample measures however, there was additional inaccuracy stemming from the nature of sample gas delivery into the CRDS analyser.

25  As discussed in studies by Gkinis et al. (2011) and Stowasser et al. (2012), stepwise changes to the inlet gas composition (as occur with discrete samples) do not give rise to correspondingly abrupt jumps in CRDS measurements, and instead result in sigmoid-shaped steps in the data (Fig. 3b). These smoothed transitions are the combined result of (i) the rate of gas replenishment in the optical cavity (Stowasser et al., 2014), (ii) partial mixing (turbulence and diffusion) of gas compositions downstream of the sample inlet (Gkinis et al., 2011), and (iii) molecular sorption and desorption on internal surfaces of the

30  cavity and inlet tubes (Friedrichs et al., 2010). Although 'response times' of CRDS instruments typically range 1 to 3 min (Picarro, 2011; Sumner et al., 2011), the actual time required for an optical cavity to completely transition to a new gas composition can be substantially longer. In testing the G2131-i, we observed remnants of previous gases persisting with asymptotical decline for as long as 40 min following very large shifts in $CO_2$ composition (e.g. $|\Delta xCO_2|$ >10000 ppm or

$|\Delta\delta^{13}C\text{-}CO_2|$ >5000 ‰). While the error caused by the residual gases may sometimes be relatively trivial, all measurements that occur prior to the cavity attaining equilibrium will experience these 'memory effects'.

In our 50 ml syringe samples, memory effects were clearly present, as evidenced by the asymptotic curvature in the data peaks (Fig. 2). This meant that reported measures of syringe samples were biased towards reference air compared to 'true' values that would be eventually determined from measurements of indeterminately large sample volumes and monitoring for asymptotic closure. Other researchers have mitigated memory effects by evacuating the optical cavity before sample introduction (Berryman et al., 2011), or through several replicate measurements (Gupta et al., 2009; Leffler and Welker, 2013). Such solutions significantly reduce sample throughput however. In this work, we elected to post-correct for reference air carry-over by calibrating our method with bottled gas standards. More specifically, we compared discrete sample measurements of gas standards against measures of the same standards directly inlet to the G2131-i for prolonged periods (>1 h). Importantly, syringe measures were not calibrated directly against gravimetric values of the standards – here we were only concerned with isolating and eliminating bias associated with the syringe sampling method and not with unaccounted inaccuracies or drift in instrument spectroscopy. This approach facilitates a more comprehensive examination of memory effects while also providing flexibility for method adaptation or applying additional error corrections for specific samples (e.g. adjusting for gas-matrix pressure broadening effects: Nara et al., 2012).

To this end, seven gravimetric gas standards were used as fixed source calibrants (0.05 to 2109 ppm $x$CO$_2$, and -27.3 to +21740 ‰ $\delta^{13}$C-CO$_2$; see Table 1; exact compositions detailed in Dickinson et al., 2017). Using such a wide range of CO$_2$ compositions served to improve overall calibration accuracy as well as demonstrating reliability and applicability of method. Direct measurements were performed by inletting the bottled standards to the G2131-i for more than one hour to ensure the absence of memory effects before taking formal measures for 10 min (ca. 460 data points; averages reported in Table 1). Next, 50 ml syringe samples of the standards were taken directly from bottles (syringe was pre-flushed several times to preclude contamination) and measured as described in Sect. 2.2 (8 samples of each standard for 56 measures in total – dataset in the Supplement). Before further analysis, due to the high $^{13}$C abundance in several gas standards, all CRDS reported CO$_2$ data were adjusted using the empirical correction described by Dickinson et al. (2017). (In testing CRDS performance with $^{13}$C-enriched CO$_2$, Dickinson et al. identified minor but unaccounted spectroscopic cross-talk in $^{12}$CO$_2$ measurements at elevated levels of $^{13}$CO$_2$, as well as logical inconsistencies in the G2131-i data output. Their correction scheme was applied to the present data as a precaution to preclude any possibility that an underlying instrument error might obfuscate the memory effects in syringe sample measures.)

The relationship between syringe and bottle measurements was established by recognising that the data peaks generated by syringe samples could be approximated by generalised logistic curves (Fig. 3b; also Gkinis et al., 2011). From this, together with a constant aliquot size for all syringe measures, we were able to predict a simple linear scaling of syringe values:

Dane Dickinson 4/10/2017 14:20

Dane Dickinson 4/10/2017 14:20

Dane Dickinson 4/10/2017 14:20

Dane Dickinson 4/10/2017 14:20

Dane Dickinson 4/10/2017 14:20

Dane Dickinson 4/10/2017 14:20

Dane Dickinson 4/10/2017 14:20

Dane Dickinson 4/10/2017 14:20

Dane Dickinson 4/10/2017 14:20

Dane Dickinson 4/10/2017 14:20

Dane Dickinson 4/10/2017 14:20

Dane Dickinson 4/10/2017 14:20

Dane Dickinson 4/10/2017 14:20

Dane Dickinson 4/10/2017 14:20

Dane Dickinson 4/10/2017 14:20

Dane Dickinson 4/10/2017 14:20

$$syringe = base + (bottle - base)/K \tag{1}$$

where *syringe* refers to the measurement value obtained for a syringe sample of a gas standard, *base* to the baseline measurement of reference air prior to sample introduction, *bottle* to the direct bottle measurement of the same standard, and *K* is a dimensionless empirical constant.

While all $CO_2$ data elements reported by the G2131-i exhibited reasonably similar sample peak geometry, the empirical constants for $^{12}CO_2$ and $^{13}CO_2$ were expected to differ due to (de)sorption and diffusion induced isotope fractionation during sample filling of the optical cavity. Further, theoretical gas mixing considerations entailed Eq. (1) would not consistently hold for $^{13}C/^{12}C$ isotope ratio data ($R_{CO_2}$) where a simultaneous change in total-$xCO_2$ also occurred. Consequently, only

10 $x^{12}CO_2$ and $x^{13}CO_2$ data were explicitly calibrated, with $R_{CO_2}$ being subsequently recalculated. (Moreover, only the dry mole fraction data of $^{12}CO_2$ and $^{13}CO_2$ were used due to the high likelihood of different transition equalisation rates for $CO_2$ and $H_2O$. For explanation of dry and wet mole fraction data see: Hoffnagle, 2015; Rella, 2010a; Rella et al., 2013.) Accordingly, the following correction formulae were derived from Eq. (1):

$$x^{12}CO_2(\text{corrected}) = x^{12}CO_2(\text{base}) + [x^{12}CO_2(\text{syringe}) - x^{12}CO_2(\text{base})] \cdot K_{C12} \tag{2}$$

15 $$x^{13}CO_2(\text{corrected}) = x^{13}CO_2(\text{base}) + [x^{13}CO_2(\text{syringe}) - x^{13}CO_2(\text{base})] \cdot K_{C13} \tag{3}$$

Total-$xCO_2$, $R_{CO_2}$, and $\delta^{13}C$-$CO_2$ data were then determined from the resulting corrected values of $x^{12}CO_2$ and $x^{13}CO_2$:

$$xCO_2 = x^{12}CO_2(\text{corrected}) + x^{13}CO_2(\text{corrected}) \tag{4}$$

$$R_{CO_2} = \frac{x^{13}CO_2(\text{corrected})}{x^{12}CO_2(\text{corrected})} \tag{5}$$

$$\delta^{13}C\text{-}CO_2 = \left[\left(\frac{R_{CO_2}}{R_{VPDB}}\right) - 1\right] \cdot 1000 \text{ ‰} \tag{6}$$

20 The correction constants, $K_{C12}$ and $K_{C13}$, were found through weighted least squares analysis (WLS) of Eqs. (2) and (3) with syringe and bottle measurements of gas standards as input data (i.e. reverse regression of Eq. 1; bottle measures substituting for the left-hand-sides of Eqs. 2 and 3). To increase statistical power, $R_{CO_2}$ and total-$xCO_2$ data from bottle measurements were also incorporated into the analysis with Eqs. (4) and (5), thereby forming an extended optimisation problem (n = 216). In a similar vein to the WLS approach used by both Dickinson et al. (2017) and Stowasser et al. (2014) for calibrating CRDS

25 measures, residuals weights were taken as the reciprocals of the individual summed variances resulting from the SDs of each syringe sample and bottle measurement (see Supplement and Table 1). The WLS solution was determined in R (version 3.2.1; R Core Team, 2015) by general purpose optimisation using the L-BFGS-B algorithm (Zhu et al., 1997) to yield the best-fit correction constants for all available $CO_2$ mole fraction and $^{13}C/^{12}C$ isotope ratio data.

**2.4 Precision and consistency tests**

CRDS precision is generally assessed by the variability in repeated measurements of a homogenous gas source (e.g. the SD of multiple 5 min analyses; Vogel et al., 2013; Wang et al., 2013). However, the internal variation in individual measurements can also be used to gauge analytic resolution (e.g. the SD of data contained in a 10 min measure – as with the bottle measurements in Sect. 2.3, also Pang et al., 2016 and Stowasser et al., 2014). For our case of 50 ml syringe samples, precision was quantified in both ways: The SD of the 30 s of CRDS data composing each individual sample (intra-sample SD, see Sect. 2.2), and as the statistical dispersion of replicate samples (inter-sample SD).

We tested method precision by repeated measurements of a systematic set of gas mixtures that spanned 
[revised manuscript text omitted]
). It is important to note that because these data were only adjusted for memory effects inherent to the discrete sample system (i.e. by Eqs. 2–6), they represent a simultaneous time-series test of instrument stability and methodological constancy. The measures averaged 1024.18 ppm in $x^{12}CO_2$ and -27.35 ‰ in $\delta^{13}C$-$CO_2$ with respective SDs of 0.50 ppm and 0.33 ‰. The latter SD is larger than the inter-sample SD found in replicate measure testing (0.15 ‰, see above), likely

5  indicating the presence of instrument drift in the data in addition to random errors of repeated syringe sampling. While the separate components of variance cannot be resolved here, moving-means (red lines in Fig. 6) show neither a sustained trend nor method discontinuity, and imply that reasonable measurement accuracy is possible under typical laboratory practices without perpetual calibration against gas standards (compare syringe sample measures against the direct bottle measurement of NA2; Fig. 6), corroborating the similar conclusion reached by Friedrichs et al. (2010). The mean of intra-sample SDs in

10 the 200 measures was 0.42 ppm for $x^{12}CO_2$ and 0.35 ‰ for $\delta^{13}C$-$CO_2$, both corresponding well to the aforementioned SDs of all measurements and the intra-sample SDs in the replicate tests. This consistency further supports our proposition that a single syringe measure and its intra-sample SD can deliver a similar (although inherently less reliable) statistical estimate to one generated through multiple sample measurements, potentially making replicate CRDS analyses unnecessary in research contexts where statistical uncertainty is not a critical consideration.

In sum, despite the short CRDS analysis period for a syringe sample (ca. 30 s), and limited number of replicates in performance testing, achieved measurement precision was excellent. With our system and G2131-i analyser, replicate sample SDs of ≤0.05 % may be expected for $^{12}CO_2$ and $^{13}CO_2$ mole fraction data, while resolution in repeated $\delta^{13}C$-$CO_2$ measurements will be ca. 0.15 ‰ at natural $^{13}C$ abundance. Moreover, to a first approximation, similar precisions can be

20 obtained from intra-sample SDs of single syringe sample measures. These results should be viewed tentatively if adapting our method to a different model of CRDS analyser however. Because observed measurement variation derives from both volatility in the discrete sampling method and noise inherent to the instrument, matching the precision reported here is unlikely with lower performance CRDS units. As a point of reference, when using the G2131-i for continuous analysis of a homogeneous source, the inter-sample SDs for sequential 30 s data segments are ca. 0.15 ‰ in $\delta^{13}C$-$CO_2$ and 0.01 % in both

25 $x^{12}CO_2$ and $x^{13}CO_2$, while intra-sample SDs are respectively ca. 0.30 ‰ and 0.025 % (Dickinson et al., 2017; also manufacturer specifications: Picarro, 2011).

Comparing to other CRDS discrete sample methods, our results improve upon the 0.3 % ($x$CO$_2$) and 0.3 ‰ ($\delta^{13}C$-$CO_2$) precision attained by Berryman et al. (2011), although this is probably due to the enhanced spectroscopic sensitivity of the

[revised manuscript text omitted]